# Morphological and Tissue Characterization with 3D Reconstruction of a 350-Year-Old Austrian *Ardea purpurea* Glacier Mummy

**DOI:** 10.3390/biology12010114

**Published:** 2023-01-11

**Authors:** Seraphin H. Unterberger, Cordula Berger, Michael Schirmer, Anton Kasper Pallua, Bettina Zelger, Georg Schäfer, Christian Kremser, Gerald Degenhart, Harald Spiegl, Simon Erler, David Putzer, Rohit Arora, Walther Parson, Johannes Dominikus Pallua

**Affiliations:** 1Material-Technology, Leopold-Franzens University Innsbruck, Technikerstraße 13, 6020 Innsbruck, Austria; 2Institute of Legal Medicine, Medical University of Innsbruck, Muellerstraße 44, 6020 Innsbruck, Austria; 3Department of Internal Medicine, Clinic II, Medical University of Innsbruck, Anichstrasse 35, 6020 Innsbruck, Austria; 4Former Institute for Computed Tomography-Neuro CT, Medical University of Innsbruck, Anichstraße 35, 6020 Innsbruck, Austria; 5Institute of Pathology, Neuropathology, and Molecular Pathology, Medical University of Innsbruck, Muellerstrasse 44, 6020 Innsbruck, Austria; 6Department of Radiology, Medical University of Innsbruck, Anichstraße 35, 6020 Innsbruck, Austria; 7WESTCAM Datentechnik GmbH, Gewerbepark 38, 6068 Mils, Austria; 8University Hospital for Orthopaedics and Traumatology, Medical University of Innsbruck, Anichstraße 35, 6020 Innsbruck, Austria; 9Forensic Science Program, The Pennsylvania State University, State College, PA 16801, USA

**Keywords:** magnetic resonance imaging, micro-computed tomography, radiocarbon dating, DNA barcoding

## Abstract

**Simple Summary:**

As glaciers disappear, animal mummies preserved in ice for centuries are released. Depending on the preservation method, residual soft tissues may differ in their biological information content. Paleoradiology, including micro-computed tomography (micro-CT) and magnetic resonance imaging (MRI), is the method of choice for the non-destructive analysis of mummies. A 350-year-old Austrian *Ardea purpurea* glacier mummy from the Öztal Alps was identified with micro-CT, MRI, histo-anatomical analyses, and DNA sequencing.

**Abstract:**

Glaciers are dwindling archives, releasing animal mummies preserved in the ice for centuries due to climate changes. As preservation varies, residual soft tissues may differently expand the biological information content of such mummies. DNA studies have proven the possibility of extracting and analyzing DNA preserved in skeletal residuals and sediments for hundreds or thousands of years. Paleoradiology is the method of choice as a non-destructive tool for analyzing mummies, including micro-computed tomography (micro-CT) and magnetic resonance imaging (MRI). Together with radiocarbon dating, histo-anatomical analyses, and DNA sequencing, these techniques were employed to identify a 350-year-old Austrian *Ardea purpurea* glacier mummy from the Ötztal Alps. Combining these techniques proved to be a robust methodological concept for collecting inaccessible information regarding the structural organization of the mummy. The variety of methodological approaches resulted in a distinct picture of the morphological patterns of the glacier animal mummy. The BLAST search in GenBank resulted in a 100% and 98.7% match in the cytb gene sequence with two entries of the species Purple heron (*Ardea purpurea*; Accession number KJ941160.1 and KJ190948.1) and a 98% match with the same species for the 16 s sequence (KJ190948.1), which was confirmed by the anatomic characteristics deduced from micro-CT and MRI.

## 1. Introduction

Glaciers release animal mummies preserved in the ice for centuries or millennia due to climate changes. These mummies are of inestimable value from an archaeological and biological point of view. Glacier animal mummies are relatively recent, as global temperature evolution has shown pronounced warming over the past 150 years. The location of such glacier animal mummies can be natural habitats, hiding places from predators, migration paths, or transport areas through updrafts [1]. Such animal mummies provide the unique opportunity to evaluate and compare different procedures for the analysis of glacier animal mummies, which can then be applied to human glacier mummies. The histological analysis of soft tissues may further expand the information content [2]. Internal organs, such as those comprising the digestive system, are often entirely decomposed. Organs may be shrunken and challenging to identify. The most oft-preserved soft tissues are those with a high collagen content, such as the dermis, muscle fasciae, and tendons [3].

The presence of skin may give essential clues regarding pathology and trauma. Good soft tissue preservation may also indicate good DNA preservation, allowing for genomic species identification [4]. Ancient DNA studies have proven the possibility of extracting and analyzing DNA preserved in skeletal remains and sediments for hundreds of thousands of years [1,2,3,4]. Such discoveries assist scientists in answering questions about extinct species and their relationships to others, including animals alive today. However, the very presence of soft tissue, especially the skin, also makes it challenging to examine the body in a non-destructive manner. Many studies focus on developing and applying non-destructive methods for analyzing mummies, for which paleoradiology is the method of choice [5]. Paleoradiologic analyses enable mummies to remain intact, protecting a valuable archaeological resource. These analyses could disclose information on the nature of the skeletal remains and the mummification process.

Generally speaking, computed tomography (CT) is the gold standard diagnostic method for mummy studies [6]. In addition, magnetic resonance imaging (MRI) has successfully been applied to ancient specimens [7]. The soft tissues that are found in mummified remains display radio-anatomical characteristics that are different from those known from clinical data. In ancient mummies, post-mortem dehydration and decomposition often lead to skin folding, and soft tissues appear radio-opaque on CT [8]. The usual lack of moisture in historical material makes MRI exceedingly tricky. MRI has successfully been applied in ancient dry soft tissues after invasive, morphology-alternating rehydration [9,10] and by using MRI settings with ultra-short echo time sequences [7,11]. However, paleoradiologic diagnostic accuracy and spatial tissue differentiation in historic mummies are not satisfactory due to tissue alterations. Therefore, the desire to achieve a high degree of diagnostic sensitivity and specificity is crucial in choosing any methodological approach in studies on glacier animal mummies. The discovery of one of the best-preserved human glacier mummies in the Ötztal Alps laid the foundations for scientific endeavors to diagnose glacier-bearing objects [12,13,14,15,16]. Due to the rarity of these findings, there is no standardized process for investigation. The handling and examination of glacier mummies are complex due to their rare occurrence and the associated lack of experience.

Micro-CT is an imaging procedure that is based on the same physical and technical bases as CT. These devices are primarily a miniaturized form of volume- or cone-beam CT scanners and can be used for non-invasive, three-dimensional investigations in preclinical research on bones, teeth, and small animals. A significant advantage of using micro-CT compared to clinical CT is a considerably higher spatial resolution with significantly better visualization of anatomical structures [17,18,19]. In vivo measurements with a spatial resolution of 10 µm are possible [20,21]. Thus, micro- and nano-CTs comprise an essential non-destructive tool to study internal structures in various disciplines, including biology [22,23,24,25,26,27], paleontology [28], geology [29], thermochronology [30], hydrology [31], soil science [32,33,34,35], materials science [36,37], and medicine [38,39,40,41,42,43].

In order to conduct a comprehensive non-destructive investigation, MRI was added. MRI provides a non-invasive tool to investigate the internal anatomy and physiology of living organisms and exploits the phenomenon of nuclear magnetic resonance. With this method, atomic nuclei exposed to a strong magnetic field absorb and reemit electromagnetic waves at characteristic frequencies, providing information on the structural and biochemical properties of the tissue [44,45]. Therefore, micro-CT and MRI analyses might offer a valuable and novel extension to conventional methods for glacier mummy research. Ideal tools for the morphological and histomorphological evaluation of mummy specimens include fixation, embedding, cutting, mounting on slides, staining, and examination using microscopic techniques such as optical, electron, and fluorescence microscopy. These methods require substantial preparation procedures and interpretation by experts. Micro-CT and MRI represent complementary tools, enabling the investigation with or without sample processing before image acquisition [46]. These imaging techniques are powerful tools, with several advantages in the structural characterization of biological systems: nearly no damage to samples occurs, and repeated scanning of the same sample is possible [47]. Moreover, imagery from micro-CT and MRI measurements can also be viewed and reviewed in 2D or 3D, and objects of interest can be segmented from the images as digital surfaces or isosurfaces to analyze complex structures [30].

A glacier mummy was found at the Gurgler Ferner on North Tyrolean territory near Ötztal, Tyrol, Austria. This study’s primary goal was to apply paleoradiological imaging in the form of micro-CT and MRI analyses, followed by radiocarbon dating and DNA analyses, to this animal glacier mummy. Thus, modern methods for investigating glacier mummies are explored for potential use in human glacier mummies.

## 2. Materials and Methods

### 2.1. Find Spot

A glacier mummy was found at 3004 m height on 3 August 2015 by Franz Scheiber and Josef Klotz in the area of the Hochwildehaus towards Hochwilde and the Annakegele at the Gurgler Ferner (degree of latitude: 46.785946, degree of longitude: 11.003878) on North Tyrolean territory near Ötztal, Tyrol, Austria (see Figure 1A). The Gurgler Ferner in Tyrol is one of the largest glaciers in the Ötztal Alps. With an area of 9.58 km², it is now the third-largest glacier in the Austrian province of Tyrol [48]. Due to the demarcation of the border, which in this area is not always oriented to the ice or watershed, smaller parts of the glacier are also located on Italian territory and are protected in the South Tyrolean nature park Texelgruppe. The Gurgler Ferner is a typical valley glacier and flows from the Gurgler ridge, which is part of the main alpine ridge, almost eight kilometers to the north into the Gurgler valley [49]. The Gurgler Ferner is embedded between the Ramolkamm with the Schalfkogel in the west and the Schwärzenkamm in the east. On the orographic right bank of the glacier lies the Hochwildehaus. In this area, on 6 August 2015, Franz Scheiber, Josef Klotz, Judith Unterberger, and Seraphin Unterberger collected the mummy parts (see Figure 1B). All parts were sealed in bags and boxes. The head was separated from the body, and the plumage was packed separately in a plastic bag with undefinable parts when recovered (see Figure 1C,D). These boxes and bags were frozen at −18 °C until the start of the investigation.

### 2.2. Micro-Computed Tomography (Micro-CT)

Micro-CT measurements were performed on a vivaCt40 and an XtremeCT II (ScancoMedical AG, Brüttisellen, Switzerland). Due to the geometrical dimensions, the corpus of the mummy was scanned only in the XtremeCT II. The mummy’s skull was further analyzed in the vivaCT40 due to its higher resolution. The settings for the XtremeCT II experiments were a 30.5 µm isotropic voxel size with a 68 kV, 1470 µA tube setting and 650 ms exposure time. The image matrix was 4096 × 4096, with a 16-bit grey-value resolution. The settings for the micro-CT experiment were a 10.5 µm isotropic voxel size with a 70 kV, 114 µA tube setting, 6500 ms exposure time, 1000 projections, and 2048 samples. The image matrix was 2048 × 2048 with a 16-bit grey-value resolution. The micro-CT data were evaluated by an experienced radiologist and summarized. The reconstructions were carried out with Analyze 14.0 (Analyze Direct Inc., Overland Park, KS, USA) software. The following two-dimensional display formats were used for the reconstructions:Planar CT slice images: reconstructed CT slice images along axial, sagittal, or coronal planes.Multi-planar reconstruction: reconstructed CT slice images along planes of freely selectable position and angle.Curved planar reconstruction: reconstructed CT slice images along planes of arbitrary orientation.

Various three-dimensional reconstructions were also used:Maximum intensity projection: the voxel with the highest intensity is displayed along a specific projection through the volume dataset.Surface Rendering: according to a defined mean value, the surface is rendered along a certain projection of the volume data set.Volume Rendering: assignment of a color value to a voxel according to its X-ray density. It is then possible to make certain regions transparent.Segmentation: semi-automatic segmentation tools such as Threshold Volume, Region Grow, and Object Extractor were used to segment the calcifications and internal organs. The procedure consists of selecting a pixel within an area called a seed. Neighboring pixels of similar density are automatically added or connected. A density threshold is chosen to capture the total volume in the pmCT layer. This process is repeated for each layer, and the total volume is automatically added.

For the comparative morphological study, 12 individual bones (sternum, coracoid, scapula, furcula, humerus, radius, ulna, carpometacarpus, pelvis, femur, tibiotarsus, and tarsometatarsus) of the postcranial skeleton were used and compared with published data [50]. The measurements were carried out as described in [50].

### 2.3. Magnetic Resonance Imaging (MRI) Data Acquisition and Processing

The MRI experiments were performed on a 3-T MR-Scanner (Siemens Magnetom Skyra, Siemens Healthineers, Erlangen, Germany) using a standard 12-channel head coil. Magnetization prepared T1 weighted ultrashort echo time imaging using PETRA (“Pointwise Encoding Time Reduction with Radial Acquisition”) was used [51] with TR = 3.32 ms, TE = 0.07 ms, TI = 1300 ms, flip angle: 6°, receive bandwidth: 400 Hz/pixel, number of radial views: 60,000, FOV: 251 mm, image matrix: 320 × 320, voxel size: 0.78 mm × 0.78 mm × 0.78 mm. In addition, a multi-slab T2-weighted turbo spin-echo sequence was acquired with TR = 6680 ms, TE = 103 ms, echo train length: 15, slice thickness: 2 mm, spacing between slices: 2.2 mm, acquisition matrix: 448 × 314, FOV: 140 mm, voxel size: 0.31 mm × 0.31 mm × 2 mm, number of slabs to cover the whole bird: 4 with 25 images per slab. Data processing and analyses were performed using Syngo.Via (Siemens Healthcare, Erlangen, Germany).

### 2.4. Sample Collection and Tissue Specimens

First, the animal glacier mummy was macroscopically examined. After using the paleoradiological methods, two independent biopsies were taken from different organs. As these did not provide conclusive results, the mummy was then sequentially sliced from caudal to cranial at 3–5 mm intervals, fixed in formalin, and embedded in paraffin as whole-mount sections according to the European standards of Biobanking CEN/TS and the ISO standards ISO 20166-1:2018, ISO 20166-2:2018, and ISO 20166-3:2018 on the pre-examination process for molecular diagnostics [52,53]. This technique assures the preservation of tissues for future histological and biomolecular analyses. Before formalin fixation, samples were taken for radiocarbon dating and DNA sequencing. Figure 2 presents the macroscopical inspection and sampling for DNA sequencing and radiocarbon dating.

### 2.5. Radiocarbon Dating

Radiocarbon dating was routinely performed at the Ion Beam Physics, ETH Zurich Laboratory. Two samples (20 mg and 200 mg) consisting of feather, skin, bone, and tissue were used. Sample treatment [54,55], reporting [56,57], and reporting ^14^C ages [58] were conducted according to the cited literature.

### 2.6. DNA Analysis

Two tissue punches (size 1.5 × 2 mm) and one bone sample from the glacier mummy were used for the DNA analysis. The tissue punches were lysed, and DNA was extracted using the EZ-1 and the MagAttract DNA kit (all Qiagen, Hilden, Germany) following the manufacturer’s recommendations.

Physical and chemical cleaning of the bone surface: The mechanical and chemical processing of the samples was performed with the necessary care required for forensically relevant samples containing only minute amounts of DNA exposed to potential superficial contamination [59,60]. One bone sample was taken from the unknown animal glacier mummy and subjected to mechanical surface cleaning with sterile scalpel blades. The sample was then bathed in sodium hypochlorite (≥4% active chlorine, Sigma Aldrich, St. Louis, MO, USA) at room temperature for 15 min, washed twice in purified water (DNA/RNA free), and rinsed in absolute ethanol for 5 min. Samples were dried in a closed laminar flow cabinet overnight, UV irradiated for 10 min (λ = 254 nm), and then powdered using a vibrating ball mill (Mixer Mill MM400, Retsch, Haan, Germany). Grinding with the ball mill was performed in cycles of 60 s, with a grinding rate of 25 Hz, followed by 60 s cooling steps. A minimum of two grinding cycles were completed, and the abrasive product was visually evaluated for homogeneity.

DNA extraction of the bone sample: The bone powder was subjected to lysis, and DNA was extracted according to the modified Dabney method as described in Xavier et al., 2021 [61].

Mitochondrial DNA typing: Both the mitochondrial (mt)DNA cytochrome b gene [62], as well as the 16 sRNA [63], were amplified and sequenced on an ABI 3500 Sequencer, and the sequences were aligned using Sequencher (GeneCodes, Ann Arbour, MI, USA). The consensus sequence was BLASTed at NCBI GenBank to recover the closest neighbors for species identification.

## 3. Results

### 3.1. Radiocarbon Dating

Data of the radiocarbon dating of the *Ardea purpurea* glacier mummy are shown in Figure 3. The sample indicates the presence of “bomb peak ^14^C” (post 1950 AD) 1 sigma range BC/AC Lower 1642 Upper 1665, 2 sigma range BC/AC Lower 1529 Upper 1799. All calibrated intervals listed below need to be taken into account. In some cases, due to the shape of the calibration curve in the region of interest, the sample’s age falls into a period when precise information about the true age range cannot be provided. Therefore, radiocarbon dating defined the mummy’s age as 350 years.

### 3.2. Morphological Analysis via Micro-CT and MRI

Before further destructive analyses, micro-CT and MRI were used with photographs of anatomical sections to study anatomy. The micro-CT and MRI scans were correlated with three photograph images of the anatomical section (see Figure 4) to identify relevant structures along the trunk from the crop to the end of the thoracoabdominal cavity. A localization image for micro-CT and MRI is shown in Figure 4, with the transverse planes and corresponding anatomical sections shown in lines. Micro-CT, MRI, and anatomical section examination revealed the presence of remains of internal organs, many of which appeared to be lytic. Brain tissue remains were not visible on the inside of the calotte. The remains of the lungs, heart, stomach, and other internal organs were also visible. The lungs are collapsed, and their outlines were still recognizable. The alveolar and bronchial structures were still clearly visible. The remains of all major muscle groups were also preserved. Fat deposits were still visible. The examination of the remaining structures revealed no pathological modifications, and the specific cause of death could not be conclusively determined. There were no signs of artificial body mummification (e.g., no puncture channels, no opening of the body cavities, no removal of organs, and no introduction of foreign material).

Three-dimensional reconstructions of the skull and the postcranial skeleton based on the micro-CT data of the *Ardea purpurea* glacier mummy were performed for comparative morphological studies on single bones. All bones were in their anatomical position and completely preserved (see Figure 5). The skull was heavily pneumatized, and the posterior skullcap was large. Prominent blood–brain conductors and the bone spur were still present. The beak was deformed but not bent (see Figure 5A,B). Based on the skull, no immediate species identification could be performed due to the severe deformations of the skull. The skeleton showed no fractures, which is demonstrated by the maximum intensity projection (Figure 5C), volume rendering bone of the body (Figure 5D), and volume rendering soft tissue of the body (Figure 5E).

As a result, most of the investigated bone elements fell within the size range of modern *Ardea purpurea*. The dimensions of the coracoid, scapula, humerus, radius, ulna, carpometacarpus, pelvis, femur, tibiotarsus, and tarsometatarsus showed the smallest measured value compared to modern *Ardea purpurea*.

The furcula, coracoid, and pelvis are described as follows in more detail.

Furcula: The characteristic of the furcula of herons is a thorn-like process on the hypocleidium, which extends dorsally at the bifurcation point of the two furcula branches. The hypocleidium bears a distinct suture bar on the caudodorsal side. The best way to distinguish species is the hypocleidium. It is more general in the *Ardea purpurea* and ends, tapering evenly, with two small bone serrations close to each other (see Figure 6).

Coracoid: Compared to other herons, *Ardea purpurea* has a long and slender coracoid with a high process scapularis and a deep concave margo lateralis. The acrocoracoid, when viewed cranially, is narrow as in *Nycticorax nycticorax*. All herons have a well-developed processus scapularis that curls medially with a broad base. In the *Ardea purpurea*, this tip is drawn more craniodorsally than in other herons. The processus lateralis is bent up in the shape of a hook; together with the edge drawing from it to the apex lateralis, the margo lateralis is an essential distinguishing feature. The edge of the small hook, viewed cranially or caudally, drops vertically to bend into a concave edge. Here, the arcuate notch extends deeper into the coracoid plate. At about two-thirds of the total length, the coracoid shaft widens towards the lateral process. The labium articulare sternale extends evenly in a slight arc from the medial margin to the apex lateralis. The asymmetrical design of the articular grooves at the anterior margin of the sternum causes a different form of the ventral end of the coracoids on both sides.

For this reason, the coracoids of both sides were measured. The ventral section of the right coracoid is slightly wider than that of the left, expressed in the measurements broad basal (BB) and broad of the facies articularis basalis (BF). Additionally, the greatest diagonal length (acrocoracoid–apex lateralis) and medial length (acrocoracoid–apex medialis) were determined (see Figure 7).

BB; broad basal = 20.67 mm.BF; broad of the facies articularis basalis = 14.23 mm.GL; greatest diagonal length (acrocoracoid–apex lateralis) = 54.10 mm.LM; length medial (acrocoracoid–apex medialis) = 51.90 mm.

Pelvis: In herons, the praeacetabular portion of the pelvis is about as long as the postacetabular portion. The pelvis of *Ardea purpurea* is small and narrow. The muscle lines further caudally than in *Ardea cinerea* and ends on a small scale of bone. The cranial tip of the crista spinalis protrudes cranially beyond the pars glutaea ossis ilium. The foramina intertransversaria medialia are less numerous and smaller. The 3D reconstruction of the pelvis in Figure 8 clearly shows taphonomic bone loss.

The sternum could not be reconstructed three-dimensionally and therefore could not be measured correctly. The comparative description of the bones and the differentiation criteria published by Kellner [50] could be confirmed using the 3D reconstructions of the bones. Based on the anatomic characteristics published by Kellner [50], the glacier mummy was determined to be an adult male *Ardea purpurea*.

### 3.3. DNA Analysis

The bone sample was successfully examined in the cytb gene region (358 bp (13,733–14,090) and 16 s region (604 bp, 1943–2545—primer sequences included, positions according to reference sequence KJ190948.1). The BLAST search in GenBank resulted in a 100% and 98.7% match of the cytb gene sequence with two entries of the species Purple heron (*Ardea purpurea*; Accession number KJ941160.1 and KJ190948.1) and a 98% match with the same species for the 16 s sequence (KJ190948.1). The two tissue samples did not result in usable sequences.

## 4. Discussion

This workup of a serendipitously found glacier mummy from the Gurgler Ferner on the North Tyrolean territory near Ötztal (Tyrol, Austria) allowed for the identification of a male *Ardea purpurea*. This identification was confirmed by DNA analyses and anatomical observations with high matches of the cytb gene sequence and the 16 s sequence using a BLAST search in GenBank. Radiocarbon dating defined the mummy’s age as 350 years. Then, paleoradiological techniques, including micro-CT and MRI, allowed for a 3D reconstruction showing the mummy’s skull and body for further comparison with previously described anatomical structures [50].

Since mummies are potentially precious relics of past times, non-invasive techniques are preferred, and CT imaging is currently the most common method. The multiple planes with optimized settings for different tissues in micro-CT and MRI images have allowed species identification based on osteoanatomical observations for about 40 years [64]. Micro-CT and MRI images allowed for the three-dimensional reconstruction, especially of the skull and pelvis for sex-specific features and of the sternum, coracoid, scapula, furcula, humerus, radius, ulna, carpometacarpus, pelvis, femur, tibiotarsus, and tarsometatarsus for bone identification, as previously shown for *Ardea purpurea* [50]. Only the sternum and parts of the pelvis were severely degraded in the glacier mummy (e.g., leaching of calcium from the bones) and could not be fully morphologically assessed. As a result of the microenvironment, demineralization may not be uniform within the skeletal system or bone. The bone may appear patchy despite being morphologically intact.

Moreover, other tissues seem to become more radio-dense (i.e., the attenuation of X-ray beams increases). In particular, this affects ligaments, fasciae, and the subcutis [3]. This may be due to the deposition of mineral salts (containing metals such as iron) in collagenous tissues. Organs may require manual segmentation in several slices due to the changes caused by water loss. Therefore, the images need to be post-processed by segmenting, delineating, and extracting specific anatomical structures.

Complementing CT images with MRI data is desirable due to the lack of contrast in soft tissues found in CT images. Mummified tissues are invisible to standard MRI techniques due to their dehydration, short T2 relaxation times, and special acquisition, and thus strategies such as ultrashort echo time sequences (UTE) have to be used [65]. The magnetic resonance effect is observed for any atomic nucleus with special magnetic properties (magnetic moment), but hydrogen nuclei (protons) in particular have very advantageous properties. Therefore, protons (i.e., hydrogen) are usually most relevant for MRI, and images show the tissue’s water content or the properties of water within the tissue. Water content is higher in living tissues than in bone and enamel, which are highly mineralized, with little water content. In addition to the short T2 relaxation times already mentioned, this also explains the reduced value of MRI in mummies. However, in the glacier mummy studied in this work, the water content was high enough to allow for MRI (Figure 5).

As a limitation of the imaging methods, there is a certain degree of subjectivity in the segmentation process. Detailed anatomical knowledge is necessary. Overall, CT and MR scanning images and 3D renderings of internal structures and tissues should not be viewed as objective and “true” representations. Visualizing the skeleton results in many points of reference, allowing for a more accessible assessment of the remains of internal organs and structures. Pathological processes can be falsely attributed to diagenetic processes and vice versa [64]. In mummy MRIs, the greatest challenge is the extensive dehydration of the tissues, since dehydrated tissues lack the hydrogen (H) in mobile water required for standard MRI signals [65]. MRI imaging is less informative than micro-CT imaging in mummies without sufficient water content. This can be circumvented by rehydrating tissues and organs [10] (but this is an invasive procedure and may not always be possible) or by applying the technique to mummies that are not entirely dehydrated, such as this glacier mummy [66]. Accordingly, reasonable images using MRI have been reported for Ötzi the Tyrolean Iceman, Lindow Man from a bog, and a corpse from a sealed medieval Korean tomb [65].

Several confounding factors, including taphonomic processes and preservation always limit the study of mummies. We estimate that over a hundred mummies have been CT-scanned and reported at this point [64]. None of these data have been synthesized into meaningful work beyond a single individual scan. Often, insufficient imaging data on ancient tissues prevents conclusions from being drawn. Future research is needed to determine the difference between antemortem and postmortem findings in micro-CT and MRI images in long-term observations to eventually produce the most accurate results with optimized segmentation procedures. Comparisons between image-based and specimen-based information are necessary. Mummified tissue biobanks with asservation of tissue specimens have already been proposed [3]. Thus, new trends in mummy research emphasize establishing guidelines and ensuring proper scientific methodology regarding analytic methods.

## 5. Conclusions

Applying the methodological concept of micro-CT and MRI imaging in combination with invasive but established techniques such as radiocarbon dating and DNA analyses can support the identification of animal species, as in the case of this glacier mummy. Three-dimensional digitization and interactive visualization of micro-CT and MRI allowed us to conduct digital autopsies and to provide a highly detailed 3D reconstruction of the *Ardea purpura* mummy.

## Figures and Tables

**Figure 1 biology-12-00114-f001:**
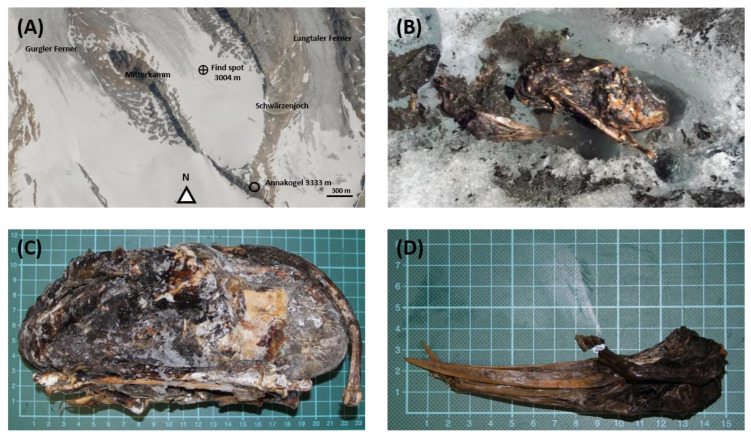
(**A**) Location of sample collection of the glacier mummy on the Gurgler Ferner using tirisMaps (https://maps.tirol.gv.at, accessed on 22 November 2022). This location is west of the sample collection of Ötzi, about 11.16 km apart. (**B**) Presentation of the glacier mummy at first finding with (**C**) focus on the corpus and the (**D**) detailed skull.

**Figure 2 biology-12-00114-f002:**
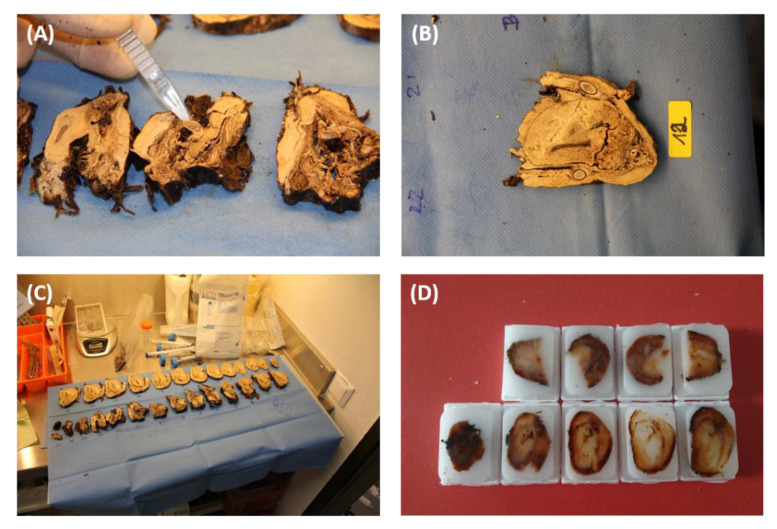
Sample collection. (**A**) Sample for DNA sequencing; (**B**) 3–5 mm slice of the mummy; (**C**) overview of the sequentially sliced mummy from caudal to cranial at 3–5 mm intervals; (**D**) paraffin-embedded tissue blocks.

**Figure 3 biology-12-00114-f003:**
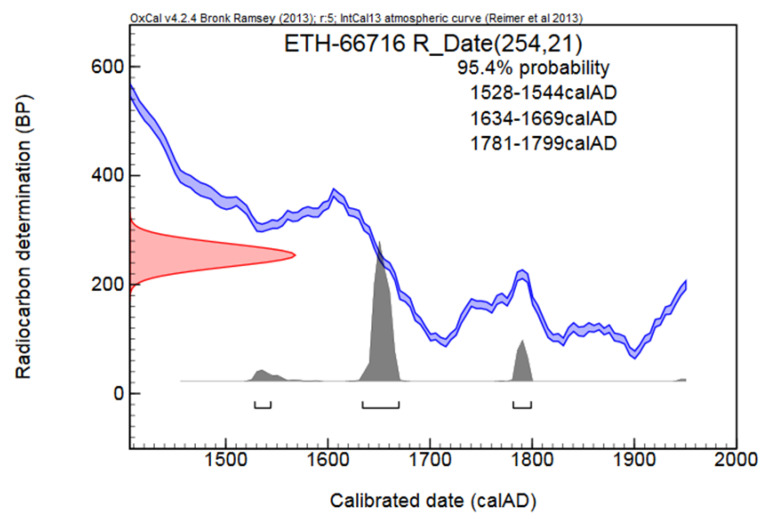
Radiocarbon dating of the *Ardea purpurea* glacier mummy. ^14^C age (BP)—delta C13 corrected radiocarbon age based on concentration of ^14^C measured in sample. BP = before present (before 1950 AD) [56,57].

**Figure 4 biology-12-00114-f004:**
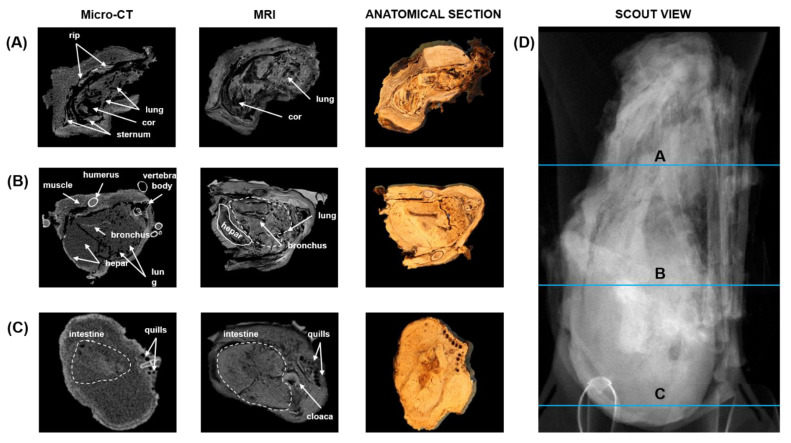
Transversal micro-CT, MRI, and anatomical section images of the trunk at different levels. (**A**) Anatomical section of the trunk at the level of the cor and lung. (**B**) Anatomical section of the trunk at the level of the hepar and lung. (**C**) Anatomical section of the trunk at the level of the cloaca and intestine. (**D**) Scout view of the trunk with lines representing the locations of different levels.

**Figure 5 biology-12-00114-f005:**
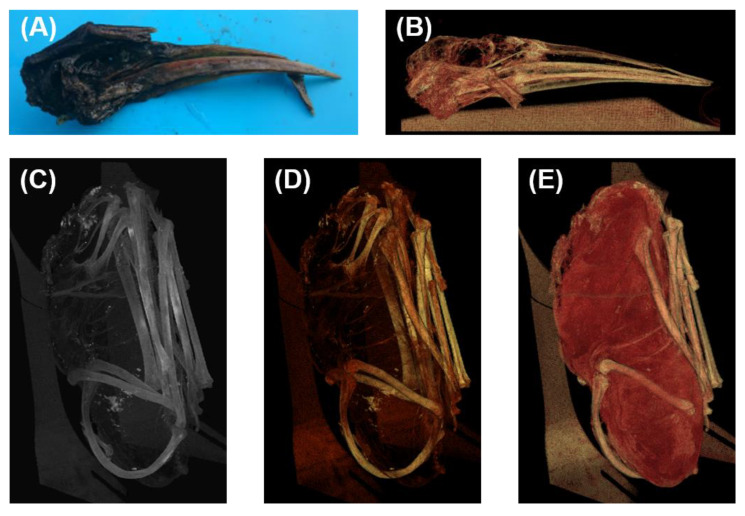
Three-dimensional reconstruction of the skull and the trunk of the *Ardea purpurea* glacier mummy. (**A**) Photograph of the skull. (**B**) Volume rendering bone of the skull. (**C**) Maximum intensity projection of the body. (**D**) Volume rendering bone of the body. (**E**) Volume rendering soft tissue of the body.

**Figure 6 biology-12-00114-f006:**
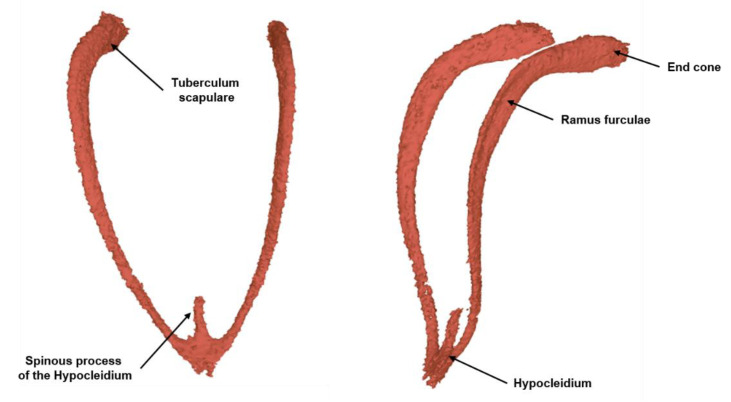
Three-dimensional reconstruction of the furcula of the *Ardea purpurea* glacier mummy.

**Figure 7 biology-12-00114-f007:**
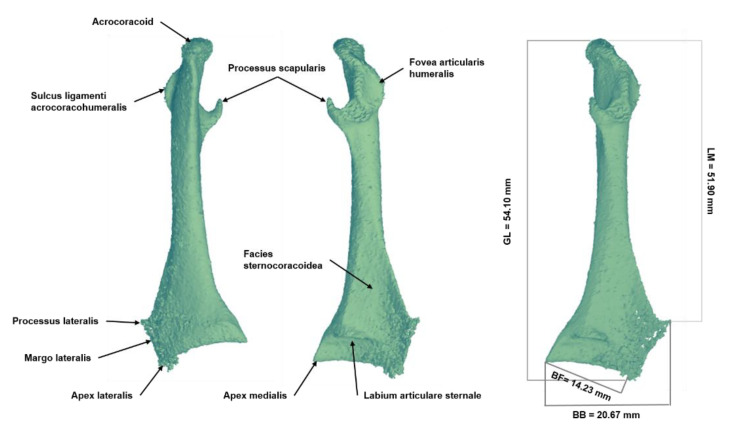
Three-dimensional reconstruction of the coracoid of the *Ardea purpurea* glacier mummy.

**Figure 8 biology-12-00114-f008:**
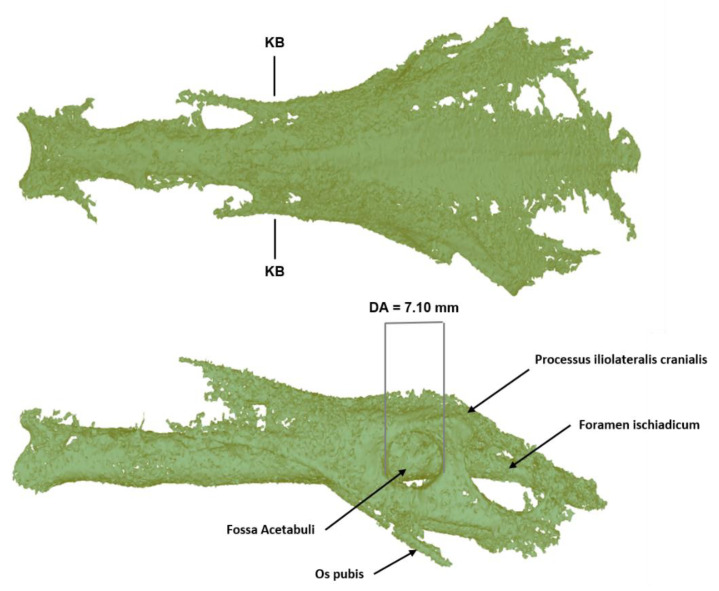
Three-dimensional reconstruction of the pelvis of the *Ardea purpurea* glacier mummy. KB: smallest width of the partes glutaeae = 12.45 mm; DA: diameter of the acetabulum (greatest distance) = 7.10 mm.

## Data Availability

Not applicable.

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
