# Peer review of "Morphological and Tissue Characterization with 3D Reconstruction of a 350-Year-Old Austrian Ardea purpurea Glacier Mummy"

_biology, 2023, doi:10.3390/biology12010114_

Round 1

Reviewer 1 Report

Dear Author,

This study is well described and interesting, but I fear it is too simplistic and that the goal of the study is unclear.

Was the goal of the study to describe this new mummy and identify it?

Or was it to provide a new standardized process for identifying mummies? (like you say in line 92?).

There is a list of 'primary goals' at line 127-129 but in the end the questions that are trying to be answered are not clear. Perhaps you may give hypotheses instead? Are you trying to provide a new protocol/methodology to identify mummies?

Moreover, the statement ' micro-CT and nano-CTs have not been widely applied in paleontological research' at line 109 is absolutely erroneous (and it is almost outrageous) because CT has been used since the 1980's on fossils and I already have in mind at least 100 papers that use CT on fossils. I think you really need to change this statement and read more literature about CT on fossils. (Where you talking specifically about mummies? Anyways, this needs to be changed).

Other important comments:

-From line 247-263, you make a description but I don't see any figures called. Please call the figures after each type of description.

-Section 3.3 Light microscopy: This section is way too short and way too simplistic for a research paper. I can see this section be 10 times bigger, please be more descriptive. You even call your workup as 'detailed' at line 363 but I would argue that this study is absolutely not detailed the way that it is written. It is quite vague on multiple aspects. You can change this by improving your introduction (stating exactly why you wrote this paper and analysed this mummy) and giving perhaps some hypotheses or research questions.

- line 374 'scince' seems mispelled

- Line 425-427 : This is remarkable, I believe this is your goal for this study, but in the end it did not deliver. Perhaps changing the introduction and then giving more clear results is key to improving the quality of this paper (I think it is too simplistic and descriptive as it is right now). But I really shouldn't be too difficult to improve.

Thank you for this interesting study nevertheless. It has the potential to become a very good paper and perhaps a milestone for the methodological future of mummy analyses.

Best wishes

Author Response

Rev 1

Dear Author,

This study is well described and interesting, but I fear it is too simplistic and that the goal of the study is unclear.

Was the goal of the study to describe this new mummy and identify it?

Or was it to provide a new standardized process for identifying mummies? (like you say in line 92?).

There is a list of 'primary goals' at line 127-129 but in the end the questions that are trying to be answered are not clear. Perhaps you may give hypotheses instead? Are you trying to provide a new protocol/methodology to identify mummies?

AW: We have specified the goals of our work.

Moreover, the statement ' micro-CT and nano-CTs have not been widely applied in paleontological research' at line 109 is absolutely erroneous (and it is almost outrageous) because CT has been used since the 1980's on fossils and I already have in mind at least 100 papers that use CT on fossils. I think you really need to change this statement and read more literature about CT on fossils. (Where you talking specifically about mummies? Anyways, this needs to be changed).

AW: For clarification we deleted this information, as it is already in the manuscript. Indeed, micro- and nano CTs are newer techniques, with only few experiences in glacier mummies.

Other important comments:

-From line 247-263, you make a description but I don't see any figures called. Please call the figures after each type of description.

AW: We added the missing information in the text.

-Section 3.3 Light microscopy: This section is way too short and way too simplistic for a research paper. I can see this section be 10 times bigger, please be more descriptive. You even call your workup as 'detailed' at line 363 but I would argue that this study is absolutely not detailed the way that it is written. It is quite vague on multiple aspects. You can change this by improving your introduction (stating exactly why you wrote this paper and analysed this mummy) and giving perhaps some hypotheses or research questions.

AW: We deleted the light microscopy part because it would go beyond the scope and would only confuse the reader. The histological results will be published with other molecular imaging techniques (SEM, EDS, IR imaging, etc...) in a further leading paper.

- line 374 'scince' seems misspelled

AW: We corrected the word.

- Line 425-427 : This is remarkable, I believe this is your goal for this study, but in the end it did not deliver. Perhaps changing the introduction and then giving more clear results is key to improving the quality of this paper (I think it is too simplistic and descriptive as it is right now). But I really shouldn't be too difficult to improve.

AW: Indeed, the primary goal of this study is the application of micro-CT and MRI analyses, followed by radiocarbon dating and DNA analyses in an animal glacier mummy. Thus we showed that digital autopsies can be performed in animal glacier mummies for potential use in human glacier mummies. We agree and reorganized the manuscript.

Thank you for this interesting study nevertheless. It has the potential to become a very good paper and perhaps a milestone for the methodological future of mummy analyses.

Best wishes

Reviewer 2 Report

In this article, Unterberger et al. describe a multi-disciplinary investigation of a 350-year-old mummified purple heron, found in the Ötztal Alps. The main result of the study was represented by the identification of the glacier mummy through morphology, radiology and DNA analysis. 

The investigation of a glacier animal mummy could have been a great opportunity to increase our understanding on the preservation of human and animal remains in cold environments, including the mummification process, tissue and biomolecule preservation and the possible presence of microorganisms.  

Unfortunately, the authors refrained from addressing these issues, but only focused on the identification of the animal species. Even worse, they completely damaged the mummy irrevocably by an unnecessary slicing of the mummy. I lack any understanding why this was done, and the authors didn’t provide any explanation for this destructive analysis. In contrary, they mentioned several times the importance of paleoradiology as the method of choice for a non-destructive analysis of mummies. I understand that histology may provide some additional information on the fine structure of bone or soft tissue, but why was it necessary to dissect the whole animal? Most likely a few punch biopsies have led to the same result. The authors do not say a word about it and, more generally, did not consider any conservatory or ethical issues at all. Even if it is not a human mummy, also a 350-year-old animal mummy should be treated as a cultural heritage. At the very least, a thorough discussion about the pros and cons of this approach would have been necessary. Based on what is outlined in the manuscript, I don’t see any good arguments that would have justified this drastic intervention. 

Apart from that, the whole outline of the paper is confusing, and it remains difficult to understand what the authors tried to achieve by applying this variety of different methods. In the end, they managed to identify the animal as mummified purple heron and provided some details on the preservation of bones, inner organs, and soft tissues. The identification of the species was already achieved by the anatomical analysis, so why was the DNA analysis done? Just to confirm it? And why did the authors not apply a genomic resp. metagenomic analysis by using next-generation sequencing, that is now a standard in the analysis of ancient human and animal remains? Such an analysis would have allowed to gain much more information on the animal species and the possibly presence of microorganism, etc. 

Moreover, the authors performed micro-CT and MRI that was compared to the anatomical sections. It appears that all three methods showed the same results on the preservation of various tissues. If so, why was it necessary to dissect the mummy, just to confirm the results of the MRI and micro-CT analysis? This could have been assumed already before damaging the mummy. It would have been more interesting to compare the various radiological techniques and evaluate their potential in the study of (glacier) mummies. Unfortunately, this wasn’t done. 

Finally, in the Discussion the authors are stating “Since mummies are ancient relics, non-invasive techniques are preferred, and CT imaging is currently the most common method” (page 4, lines 371-372). I fully agree with that, but the authors lead their own study to absurdity. They fail to provide any good reason for slicing the whole mummy into small pieces. Maybe one of the reasons for the failure of this study, is that the authors did not include a conservation expert or somebody with a strong background in cultural heritage. At least, I would have liked to see a specialist in animal mummies or bones involved in the study, who could have assisted not only in the identification of the species, but maybe could have added further information on the importance of the finding regarding the presence of these animals in the Alps or even the evolutionary history. 

Taken together, the study is more than disappointing and could serve as an example on how not to conduct such a work. The goal should not be to apply many different methods and see what comes out, as in this case just the identification of the species. Instead, such a study should have a clear goal, a clear planning of what methods should be used and an evaluation and thorough discussion of the results. Moreover, as also stated by the authors, the damaging of the mummy should be kept to a minimum and any invasive techniques should be well justified. 

Based on my evaluation, I clearly recommend to decline this manuscript. As the damages of the mummy are irreversible and the whole concept of the study is misdirected, I don’t think that any kind of (major) revision would be enough to turn this work into a publishable form. It is very difficult for me to imagine in which form this work could be published at all. A solution could be a clear and open debate on the ethical backgrounds and the application of different methods in mummy research.

Author Response

Rev 2

In this article, Unterberger et al. describe a multi-disciplinary investigation of a 350-year-old mummified purple heron, found in the Ötztal Alps. The main result of the study was represented by the identification of the glacier mummy through morphology, radiology and DNA analysis.

The investigation of a glacier animal mummy could have been a great opportunity to increase our understanding on the preservation of human and animal remains in cold environments, including the mummification process, tissue and biomolecule preservation and the possible presence of microorganisms.

Unfortunately, the authors refrained from addressing these issues, but only focused on the identification of the animal species. Even worse, they completely damaged the mummy irrevocably by an unnecessary slicing of the mummy. I lack any understanding why this was done, and the authors didn’t provide any explanation for this destructive analysis. In contrary, they mentioned several times the importance of paleoradiology as the method of choice for a non-destructive analysis of mummies. I understand that histology may provide some additional information on the fine structure of bone or soft tissue, but why was it necessary to dissect the whole animal? Most likely a few punch biopsies have led to the same result. The authors do not say a word about it and, more generally, did not consider any conservatory or ethical issues at all. Even if it is not a human mummy, also a 350-year-old animal mummy should be treated as a cultural heritage. At the very least, a thorough discussion about the pros and cons of this approach would have been necessary. Based on what is outlined in the manuscript, I don’t see any good arguments that would have justified this drastic intervention.

Apart from that, the whole outline of the paper is confusing, and it remains difficult to understand what the authors tried to achieve by applying this variety of different methods. In the end, they managed to identify the animal as mummified purple heron and provided some details on the preservation of bones, inner organs, and soft tissues. The identification of the species was already achieved by the anatomical analysis, so why was the DNA analysis done? Just to confirm it? And why did the authors not apply a genomic resp. metagenomic analysis by using next-generation sequencing, that is now a standard in the analysis of ancient human and animal remains? Such an analysis would have allowed to gain much more information on the animal species and the possibly presence of microorganism, etc.

Moreover, the authors performed micro-CT and MRI that was compared to the anatomical sections. It appears that all three methods showed the same results on the preservation of various tissues. If so, why was it necessary to dissect the mummy, just to confirm the results of the MRI and micro-CT analysis? This could have been assumed already before damaging the mummy. It would have been more interesting to compare the various radiological techniques and evaluate their potential in the study of (glacier) mummies. Unfortunately, this wasn’t done.

Finally, in the Discussion the authors are stating “Since mummies are ancient relics, non-invasive techniques are preferred, and CT imaging is currently the most common method” (page 4, lines 371-372). I fully agree with that, but the authors lead their own study to absurdity. They fail to provide any good reason for slicing the whole mummy into small pieces. Maybe one of the reasons for the failure of this study, is that the authors did not include a conservation expert or somebody with a strong background in cultural heritage. At least, I would have liked to see a specialist in animal mummies or bones involved in the study, who could have assisted not only in the identification of the species, but maybe could have added further information on the importance of the finding regarding the presence of these animals in the Alps or even the evolutionary history.

Taken together, the study is more than disappointing and could serve as an example on how not to conduct such a work. The goal should not be to apply many different methods and see what comes out, as in this case just the identification of the species. Instead, such a study should have a clear goal, a clear planning of what methods should be used and an evaluation and thorough discussion of the results. Moreover, as also stated by the authors, the damaging of the mummy should be kept to a minimum and any invasive techniques should be well justified.

Based on my evaluation, I clearly recommend to decline this manuscript. As the damages of the mummy are irreversible and the whole concept of the study is misdirected, I don’t think that any kind of (major) revision would be enough to turn this work into a publishable form. It is very difficult for me to imagine in which form this work could be published at all. A solution could be a clear and open debate on the ethical backgrounds and the application of different methods in mummy research.

AW: We fully acknowledge the concerns of reviewer 2. Indeed, because of the manuscript's length and readability, we did not describe the results of 2 independent biopsies, which were inconclusive concerning the DNA analyses. Therefore we applied the procedure recommended by the European biobanks to provide the results presented in this manuscript and assure future studies using histological and biomolecular analyses. This process is standardized for human pathological and forensic investigations. This issue has now been added with references to the manuscript. Taking together the reviewers’ suggestions on this issue, we deleted the light microscopy part not to go beyond the scope of this manuscript. For the urpose of this manuscript with identification of the animal, genomic resp. metagenomic analysis by using next-generation sequencing would have provided additional informations for other purposes but was considered as not justified for this work, even if these new methods are now available also for the analysis of ancient remains.

Reviewer 3 Report

Dear authors and editors, 

I have read over the above article, which was sent to me for review. Although it certainly has merit, and I support its eventual publication, here I suggest acceptance to be subordinate to the implementation of major revisions. In fact, the present version of the manuscript does not do full justice to the many potentialities of the work it describes, and the presentation is poorly organized and incomplete, lacking some crucial information/discussions that I'd expect in a palaeobiological paper.

My main comments are as follows:

0) This is comment n° 0 as it regards the very "nature" of the paper. In fact, though submitted for publication in the "Paleontology in the 21st Century" special issue, by presenting the methods used for imaging the glacier mummy of a XVII-century bird belonging to an extant species, the present manuscript is by no means palaeontological! It would become a contribution of interests to palaeontologists if it manages to show how the application of the described imaging techniques could aid in interpreting fossil mummified tissues. What is a "fossil" may sometimes be a matter of taste, but a 350-year-old glacier mummy does not represent a fossil (as far as my comprehension of the object of study of paleontology). 

1) The Introduction is loosely structured. The advantages and disadvantages of the various imaging techniques (and most prominently MRI) are addressed repeatedly in various paragraphs of the Intro. The object of investigation of your study, the bird mummy, is only cursorily mentioned in the midst of the Introduction - it should rather be placed at the end thereof. My personal impression is that the Intro may be substantially re-organized (and shortened). 

2) It is weird to read nothing about the precise geographic whereabouts of the specimen until the incipit of the Results. This information should not be treated as a result, but rather as part of the Methods/Intro. In addition, is there anything more that could be reported on the finding circumstances, such as GPS geographic coordinates, location within the glacier body, date of collection, etc...? All these data, if available, should be published.

3) There is a major issue with table 1. Here, the bones are described in comparative terms (smaller, thinner, etc...). What is the term of comparison for these bone descriptions?

4) Many of the internal organs and anatomical districts that you have labelled in the figures have not been properly described in the main text, and that's a pity. I understand that your aim is mostly methodological; however, the preservation of soft tissue-organs like the intestine for 300+ years is certainly not common, and some further description may prove interesting to your readership.

5) The Discussion is mostly a "reboot" of issues that have largely been reported in the Introduction. A revised Discussion may focus on investigating some taphonomic aspects (which are, at present, mostly neglected) as well as on showing how the application of the described imaging techniques could aid in interpreting fossil mummified tissues (see comment n°0 above).

6) Though mostly correct, the English text may be locally improved. I am not the best person to comment on this issue as I come from a romance-speaking country. Nonetheless, in the attached PDF file I have left many suggestions for improving the word flow (especially in the Intro). The word choice in the bone descriptions may also be changed to better conform to anatomical literature.

Many other minor suggestions are detailed in the attached PDF file.

Best wishes,
the reviewer

Author Response

Rev 3

Dear authors and editors,

I have read over the above article, which was sent to me for review. Although it certainly has merit, and I support its eventual publication, here I suggest acceptance to be subordinate to the implementation of major revisions. In fact, the present version of the manuscript does not do full justice to the many potentialities of the work it describes, and the presentation is poorly organized and incomplete, lacking some crucial information/discussions that I'd expect in a palaeobiological paper.

My main comments are as follows:

0) This is comment n° 0 as it regards the very "nature" of the paper. In fact, though submitted for publication in the "Paleontology in the 21st Century" special issue, by presenting the methods used for imaging the glacier mummy of a XVII-century bird belonging to an extant species, the present manuscript is by no means palaeontological! It would become a contribution of interests to palaeontologists if it manages to show how the application of the described imaging techniques could aid in interpreting fossil mummified tissues. What is a "fossil" may sometimes be a matter of taste, but a 350-year-old glacier mummy does not represent a fossil (as far as my comprehension of the object of study of paleontology).

AW: Indeed, this work is kind of applied paleontology, using different methods from paleontology in combination with standardized forensic and pathological techniques in an animal glacier mummy to gain experiences for future studies in human glacier mummies. We think that this is an interesting issue and therefore asked for the interest of such a work to be published in this special issue, which was agreed. We tried to explain this goal of the work in the manuscript, with multiple changes as proposed by the different reviewers.

1) The Introduction is loosely structured. The advantages and disadvantages of the various imaging techniques (and most prominently MRI) are addressed repeatedly in various paragraphs of the Intro. The object of investigation of your study, the bird mummy, is only cursorily mentioned in the midst of the Introduction - it should rather be placed at the end thereof. My personal impression is that the Intro may be substantially re-organized (and shortened).

AW: We agree and reorganized the introduction.

2) It is weird to read nothing about the precise geographic whereabouts of the specimen until the incipit of the Results. This information should not be treated as a result, but rather as part of the Methods/Intro. In addition, is there anything more that could be reported on the finding circumstances, such as GPS geographic coordinates, location within the glacier body, date of collection, etc...? All these data, if available, should be published.

AW: We added the missing information and transferred this section to the method part. The locations with detailed GPS data are given, too.

3) There is a major issue with table 1. Here, the bones are described in comparative terms (smaller, thinner, etc...). What is the term of comparison for these bone descriptions?

AW: We deleted this information.

4) Many of the internal organs and anatomical districts that you have labelled in the figures have not been properly described in the main text, and that's a pity. I understand that your aim is mostly methodological; however, the preservation of soft tissue-organs like the intestine for 300+ years is certainly not common, and some further description may prove interesting to your readership.

AW: We agree to this reviewer, that tissue-organs would be interesting, however, as proposed we deleted the light microscopical data and further descriptions from this manuscript.

5) The Discussion is mostly a "reboot" of issues that have largely been reported in the Introduction. A revised Discussion may focus on investigating some taphonomic aspects (which are, at present, mostly neglected) as well as on showing how the application of the described imaging techniques could aid in interpreting fossil mummified tissues (see comment n°0 above).

AW: We agree and reorganized and changed the discussion as suggested by the reviewers.

6) Though mostly correct, the English text may be locally improved. I am not the best person to comment on this issue as I come from a romance-speaking country. Nonetheless, in the attached PDF file I have left many suggestions for improving the word flow (especially in the Intro). The word choice in the bone descriptions may also be changed to better conform to anatomical literature.

AW: We have incorporated all suggestions for English improvement.

Many other minor suggestions are detailed in the attached PDF file.

Best wishes,

the reviewer

Reviewer 4 Report

Wonderful job on this paper! It's great to see the inclusion of new techniques for the study of mummified glacier specimens. I have no major concerns with the scientific content of the manuscript, and I believe it will be of interest to many paleontologists.

Author Response

Wonderful job on this paper! It's great to see the inclusion of new techniques for the study of mummified glacier specimens. I have no major concerns with the scientific content of the manuscript, and I believe it will be of interest to many paleontologists.

AW: Thank you for your review

Round 2

Reviewer 2 Report

The reviewers did not (fully) consider my comments and I only see little improvement of the manuscript. In particular, they did not comment on my concerns regarding ethical issues and the fact that they completely damaged the mummy. The only answer was: "Therefore we applied the procedure recommended by the European biobanks to provide the results presented in this manuscript and assure future studies using histological and biomolecular analyses. This process is standardized for human pathological and forensic investigations." The statement is misleading as they are dealing with an animal mummy and not a human pathological or forensic case.

I don't understand why the authors refuse to consider and discuss their invasive approach. It is not about a pathological or forensic case, but a rare finding that deserves to be conserved and studied in a reasonable way.

The authors have deleted the light microscopy part, which shows that this investigation has not provided important additional information. For the genetic analysis, that has only confirmed the identification, a small bone biopsy would have been sufficient. Therefore, it still remains incomprehensible why the mummy was completely dissected. 

I would strongly urge the authors to make a clear statement about the ethics of their approach and the motivation for the highly invasive approach in their study. 

Author Response

AW: A clear statement about the animal ethics relevant for out study is now added to the manuscript. We asked the Ethics Committee and the Animal Welfare Committee of the Medical University of Innsbruck about the concerns of Rev 2 and received the following answer:

Since the animal was already no longer alive at the time of the scientific investigation, and thus there was no risk of pain, suffering or harm to the animal as a result of the study, the Animal Welfare Board is unfortunately not responsible in this case.

As proposed we then asked the Advisory Board on Ethical Issues in Scientific Research of the Leopold Franzens University of Innsruck. This institution answered as following: The ethical question was discussed with a representatives of the animal protection committee of the Leopold Franzens University and with the chairperson of the ethics advisory board. In addition, colleagues from the University of Veterinary Medicine in Vienna were contacted. All of them concluded that there are no known legal or ethical standards for treating animal mummies.

We now included both answers into the revised version of the manuscript as an ethical statement at the end of the manuscript: “For such studies without risk of pain for the animal, neither the Ethics Committee nor the Animal Welfare Committee of the Medical University of Innsbruck considered themselves responsible for this project. The Advisory Board on Ethical Issues in Scientific Research and representatives of the animal protection committee of the Leopold Franzens University as well as colleagues from the University of Veterinary Medicine in Vienna further concluded that there are no known legal or ethical standards for projects on animal mummies in this country.”

As mentioned already in the methods section, this animal was therefore prepared according to the European standards of human Biobanking CEN/TS and the ISO standards ISO 20166-1:2018, ISO 20166-2:2018 and ISO 20166-3:2018 for human molecular diagnostics.

Reviewer 3 Report

I see that most of the reviewer’s proposal have been addressed by the authors, making the paper essentially suitable for acceptance. Before publication, I’d suggest the authors and editors to pay attention to the quality of the English text. 

Author Response

We used the Grammarly software (https://www.grammarly.com) to improve the quality of the English text.

Reviewer 4 Report

The authors made all the suggested changes, so the manuscript is ready to be accepted. Congratulations on this paper!

Author Response

AW: Thank you for this comment.

Round 3

Reviewer 2 Report

No additional comments.

Author Response

English language and style are spell checked by MDPI English editing.